# Thinking Two Moves Ahead: Anticipating Other Users Improves Backdoor Attacks in Federated Learning

## Abstract

Federated learning is particularly susceptible to model poisoning and backdoor attacks because individual users have direct control over the training data and model updates. At the same time, the attack power of an individual user is limited because their updates are quickly drowned out by those of many other users. Existing attacks do not account for future behaviors of other users, and thus require many sequential updates and their effects are quickly erased. We propose an attack that anticipates and accounts for the entire federated learning pipeline, including behaviors of other clients, and ensures that backdoors are effective quickly and persist even after multiple rounds of community updates. We show that this new attack is effective in realistic scenarios where the attacker only contributes to a small fraction of randomly sampled rounds and demonstrate this attack on image classification, next-word prediction, and sentiment analysis.

## 1 Introduction

When training models on private information, it is desirable to choose a learning paradigm that does not require stockpiling user data in a central location. Federated learning (Konečný et al., 2015; McMahan et al., 2017b) achieves this goal by offloading the work of model training and storage to remote devices that do not directly share data with the central server. Each user device instead receives the current state of the model from the central server, computes local updates based on user data, and then returns only the updated model to the server.

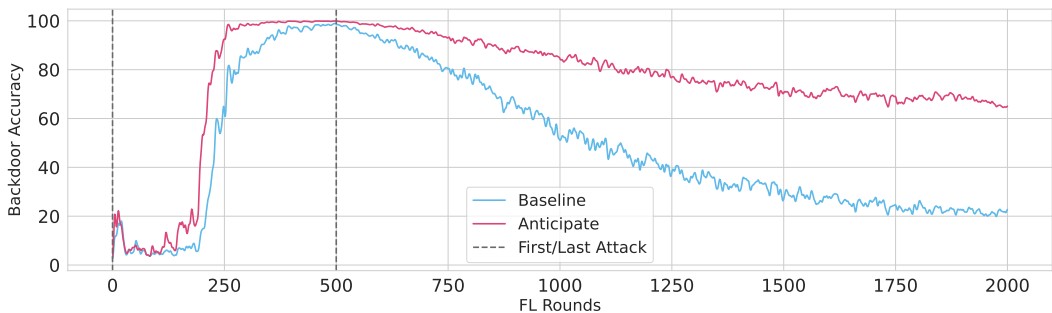

Figure 1: Our method, `Anticipate`, reaches $100\%$ backdoor accuracy faster than the baseline in the setting of 100 random attacks in the first 500 rounds. Moreover, after the window of attack passes, the attack decays much slower than the baseline. At the end of federated training, our attack still has backdoor accuracy of $60\%$, while the baseline maintains just $20\%$. Overall, only 100 out of a total of 20k contributions are malicious.

Unfortunately, by placing responsibility for model updates in the handle of many anonymous users, federated learning also opens up model training to a range of malicious attacks (Bagdasaryan et al., 2019; Kairouz et al., 2021). In *model poisoning* attacks (Biggio & Roli, 2018; Bhagoji et al., 2019),

a user sends malicious updates to the central server to alter behavior of the model. For example in language modeling, backdoor attacks could modify the behavior of the final model to misrepresent specific facts, attach negative sentiment to certain groups, change behavior in edge cases, but also attach false advertising and spam to certain key phrases.

In practical applications, however, the real threat posed by such attacks is debated (Sun et al., 2019b; Wang et al., 2020; Shejwalkar et al., 2021). Usually only a small fraction of users are presumed to be malicious, and their impact on the final model can be small, especially when the contributions of each user are limited by norm-bounding (Sun et al., 2019b). Attacks as described in Bagdasaryan & Shmatikov (2021) further require successive attacks over numerous sequential rounds of training. This is not realistic in normal cross-device applications (Bonawitz et al., 2019; Hard et al., 2019) where users are randomly selected in each round from a larger pool, making it exceedingly unlikely that any attacker or even group of attackers will be able to contribute to more than a fraction of the total rounds of training. Model updates that are limited in this way are immediately less effective, as even strong backdoor attacks can be wiped away and replaced by subsequent updates from many benign users Sun et al. (2019b); Shejwalkar et al. (2021).

In this work we set out to discover whether strong attacks are possible in these more realistic scenarios. We make the key observation that previous attack algorithms such as described in Bagdasaryan et al. (2019); Wang et al. (2020); Zhou et al. (2021) only consider the immediate effects of a model update, and ignore the downstream impacts of updates from benign users. We show that, by modeling these future updates, a savvy attacker can update model parameters in a way that is unlikely to be over-written or undone by benign users. By backpropagating through simulated future updates, our proposed attack directly optimizes a malicious update to maximize its permanence. Using both vision and language tasks, and under a realistic threat model where attack opportunities are rare, we see that these novel attacks become operational after fewer attack opportunities than baseline methods, and remain active for much longer after the attack has passed as shown in Figure 1.

## 2 BACKGROUND

Federated Learning systems have been described in a series of studies and a variety of protocols. In this work, we focus on mainly on *federated averaging* (*fedAVG*) as proposed in McMahan et al. (2017b) and implemented in a range of recent system designs (Bonawitz et al., 2019; Paulik et al., 2021; Dimitriadis et al., 2022), but the attack we describe can be extended to other algorithms. In *fedAVG*, the server sends the current state of the model $\theta_i$ to all users selected for the next round of training. Each user then computed an updated local model through several iterations, for example via local SGD. The $u$-th local user has data $D$ which is partitioned into batches $D_u$ and then, starting from the global model, their local model is updated for $m$ steps based on the training objective $\mathcal{L}$:

$$\theta_{i+1,j+1}^u = \theta_{i,j}^u - \tau \nabla \mathcal{L}(D_u, \theta_{i,j}^u), \quad \text{for } j = 1, \dots, m. \tag{1}$$

The updated models $\theta_{i+1,m}^u$ from each user are returned to the server which computes a new central state by averaging:

$$\theta_{i+1} = \frac{1}{n} \sum_{u=1}^{n} \theta_{i+1,m}^u. \tag{2}$$

We will later summarize this procedure that depends on a group of users $U_i$ in the $i$-th round as $\theta_{i+1} = F_{\text{avg}}(U_i, \theta_i)$.

Optionally, the average can be reweighted based on the amount of data controlled by each user (Bonawitz et al., 2017), however this is unsafe without further precautions, as an attacker could overweight their own contributions such that we only consider unweighted averages in this work. Federated Averaging is further safeguarded against malicious users by the use of norm-bounding. Each updated model $\theta_{i,u}$ is projected onto an $||\theta_{i,u}||_p \leq C$, for some clip value $C$ so that no user update can dominate the average.

Norm-bounding is necessary to defend against *model replacment* attacks described in Bagdasaryan et al. (2019) and Bhagoji et al. (2019) which send malicious updates with extreme magnitudes that overpower updates from benign users. Once norm-bounding is in place as a defense though, the potential threat posed by malicious attacks remains debated. We summarize a few related areas of research, before returning to this question:

**Adversarial Machine Learning**
The attacks investigated in this paper are a special case of train-time adversarial attacks against machine learning systems (Biggio et al., 2012; Cinà et al., 2022). The federated learning scenario is naturally an *online*, *white-box* scenario. The attack happens *online*, while the model is training, and can adapt to the current state of training. The attack is also *white-box* as all users have knowledge of model architecture and local training hyperparameters.

**Train-time Attacks**
In this work we are interested in backdoor attacks, also refered to as targeted attacks, which form a subset of *model integrity* attacks (Barreno et al., 2010). These attacks generally attempt to incorporate malicious behavior into a model without modifying its apparent performance on test data. In the simplest case, malicious behavior could be an image classification model that misclassifies images marked with a special patch. These attacks are in contrast to *model availability* attacks which aim to undermine model performance on all hold-out data. Availability attacks are generally considered infeasible in large-scale federated learning systems when norm-bounding is employed (Shejwalkar et al., 2021), given that malicious users likely form only a minority of all users.

**Data Poisoning**
The model poisoning attacks described above are closely related to *data poisoning* attacks against centralized training (Goldblum et al., 2020). The idea of anticipating future updates has been investigated in some works on data poisoning (Muñoz-González et al., 2017; Huang et al., 2020) where it arises as approximation of the bilevel data poisoning objective. These attacks optimize a set of poisoned datapoints by differentiating through several steps of the expected SGD update that the the central server would perform on this data. However, for data poisoning, the attacker is unaware of the model state used by the server, cannot optimize their attack for each round of training, and has only approximate knowledge of model architecture and hyperparameters. These complications lead Huang et al. (2020) to construct a large ensemble of model states trained to different stages to approximate missing knowledge.

## 3 CAN YOU BACKDOOR FEDERATED LEARNING?

Backdoor attacks against federated learning have been described in Bagdasaryan et al. (2019). The attacker uses local data and their malicious objective to create their own replacement model, scales this replacement model to the largest scale allowed by the server's norm-bounding rule and sends it. However, as discussed in Sun et al. (2019b), for a more realistic number of malicious users and randomly occurring attacks, backdoor success is much smaller, especially against stringent norm-bounding. Wang et al. (2020) note that backdoor success is high in edge cases not seen in training and that backdoors that attack "rare" samples (such as only airplanes in a specific color in images, or a specific sentence in text) can be much more successful, as other users do not influence these predictions significantly. A number of variants of this attack exist (Costa et al., 2021; Pang et al., 2021; Fang et al., 2020; Baruch et al., 2019; Xie et al., 2019; Datta et al., 2021; Yoo & Kwak, 2022; Zhang et al., 2019; Sun et al., 2022; Shen et al., 2020), for example allowing for collusion between multiple users or generating additional data for the attacker. In this work we will focus broadly on the threat model of Bagdasaryan et al. (2019); Wang et al. (2020).

**Threat Model** We assume a federated learning protocol running with multiple users, attacked by online white-box model poisoning. The server orchestrates federated averaging with norm-bounding. At each attack opportunity, the attack controls only a single user and only has knowledge about the local data from this user. The attacker has full control over the model update that will be returned to the server and can optimize this model freely. As a participating user in FL, the attacker is also aware of the number of local steps and local learning rate that users are expected to use. We will discuss two variations of this threat model with different attack opportunities. 1) An attacker opportunity is provided every round during a limited time window as in Bagdasaryan et al. (2019). 2) Only a limited number of attack opportunities arise randomly during a limited time window as discussed in Sun et al. (2019b).

We believe this threat model with random attack opportunities is a natural step towards the evaluation of risks caused by backdoor attacks in more realistic systems. We do restrict the defense to only norm-bounding and explore a worst-case attack against this scenario. As argued in Sun et al.

(2019b), norm-bounding is thought to be sufficient to prevent these attacks. We acknowledge that other defenses exist, see overviews in Wang et al. (2022) and Qiu et al. (2022), yet the proposed attack is designed to be used against norm-bounded FL systems and we verify in Appendix A.5 that it does not break other defenses. We focus on norm bounding because it is a key defense that is widely adopted in industrial implementations of federated learning (Bonawitz et al., 2019; Paulik et al., 2021; Dimitriadis et al., 2022).

## 4 ATTACKS WITH END-TO-END OPTIMIZATION

### 4.1 BASELINE

As describe by Gu et al. (2017); Bagdasaryan et al. (2019), suppose an attacker holds $N$ clean data points, $D_c = \{x_i^c, y_i^c\}_{i=1}^N$, and $M$ backdoored data points, $D_b = \{x_i^b, y_i^b\}_{i=1}^M$, where $x_i^b$ could be an input with a special patch or edge-case example (Wang et al., 2020), and $y^b$ is an attacker-chosen prediction. The goal of the attacker is to train a malicious model that predicts $y^b$ when it sees a backdoored input, and to push this behavior to the central model. The attacker can optimize their malicious objective $\mathcal{L}_{\text{adv}}$ directly to identify backdoored parameters:

$$\theta^* = \underset{\theta}{\arg\min} \, \mathcal{L}_{\text{adv}}(D_b, \theta) \tag{3}$$

where $\mathcal{L}_{\text{adv}}$ is the loss function of the task, $\theta$ are the weights of the local model. Some attacks such as Bhagoji et al. (2019) also include an additional term that enforces that model performance on local clean data remains good, when measured by the original objective $\mathcal{L}$:

$$\theta^* = \underset{\theta}{\arg\min} \, \mathcal{L}_{\text{adv}}(D_b, \theta) + \mathcal{L}(D_c, \theta). \tag{4}$$

The update is then scaled to the maximal value allowed by norm-bounding and sent to the server. This baseline encompasses the attacks proposed in Xie et al. (2019) and Bhagoji et al. (2019) in the investigated setting.

### 4.2 ANTICIPATING OTHER USERS

This baseline attack can be understood as a greedy objective which optimizes the effect of the backdoor only for the current stage of training and assumes that the impact of other users is negligible after scaling. We show that a stronger attack *anticipates* and involves the benign users' contributions in current and several future rounds during the backdoor optimization. The optimal malicious update sent by the attacker should be chosen so that it is optimal even if the update is averaged with the contributions of other users and then used for several further rounds of training to which the attacker has no access. We pose this criteria as a loss function to be optimized. Intuitively, this allows the attack to optimally select which parts of the model update to modify, and to estimate and avoid which parts would be overwritten by other users.

Formally, with $n$ users per round, suppose an attacker wants to `anticipate` $k$ steps (in the following we will use this keyword to denote the whole attack pipeline). Then, given the current local model, $\theta_0$, the objective of the attacker is simply to compute the adversarial objective in Equation (3), but optimize it not directly for $\theta$, but instead future $\theta_i, 1 \leq i \leq k$, which depends implicitly on the attacker's contribution.

To make this precise, we move through all steps now. Denote the model update that the attacker contributes by $\theta_{\text{mal}}$. In the next round following this contribution, the other users $U_0$ will themselves contribute updates $\theta_{0,u}$. Both are averaged and result in

$$\theta_1 = \frac{\theta_{\text{mal}} + \sum_j^{n-1} \theta_{0,j}}{n}, \tag{5}$$

where $\theta_1$ now depends on $\theta_{\text{mal}}$. Then, $k-1$ more rounds follow in which the attacker does not contribute, but where new users $U_i$ contribute:

$$\theta_{i+1} = F_{\text{avg}}(U_i, \theta_i). \tag{6}$$

Finally, $\theta^k$ still depends implicitly on the malicious contribution $\theta_{\text{mal}}$. As such, an omniscient attacker could then optimize

$$\theta^* = \underset{\theta_{\text{mal}}}{\arg\min} \sum_{i=1}^{k} \mathcal{L}_{\text{adv}}(D_b, \theta_i(\theta_{\text{mal}})) + \mathcal{L}(D_c, \theta_i(\theta_{\text{mal}})), \qquad (7)$$

differentiating the resulting graph of $\mathcal{L}_{\text{adv}}$ with respect to $\theta_{\text{mal}}$ and compute the gradient direction in which $\theta_{\text{mal}}$ should be updated to improve the effect of the backdoor.

However, in practice, this optimization problem is intractable. First, the attacker is unaware of the exact private data of other users in future rounds. Meanwhile, involving the full group of all users $U_i$ in the intermediate federated learning round makes the problem unsolvable for limited compute resources, given that each call to $F_{\text{avg}}$ contains many local update steps for each user which each depend on $\theta_{\text{mal}}$. Therefore, *we stochastically sample the full optimization problem*: First, we decide to model only a subset of users $n' < n$ and then randomly sample a single batch of data for each local update in each round, from the attackers own data source $D_c$. Based on this data, the attacker can then recompute the local update steps for this limited group of users and in this way stochastically approximate the real contributions from other users

---

**Algorithm 1** `Anticipate` Algorithm

---

1: **Input:** Global model $\theta_0$, batch size $b$, number of modeled users per round $n'$, future updates anticipated: $k$, update steps $m'$, the attacker $A$ owns a set of clean data $D_c$ and backdoor data $D_b$.
2: $\theta_{\text{mal}}$ = Initialize from $\theta_0$
3: **for** $0, ..., m' - 1$ **do**
4:     **for** $i = 0, ..., k - 1$ **do**
5:         Model a group of users $U_i$:
6:         **for** $u = 0, ..., n' - 1$ **do**
7:           $U_{i,u}$ = Sample $b$ data points from $D_c$
8:         Run one round of federated averaging:
9:         **if** $i == 0$ **then**
10:           $\theta_{i+1} = F_{\text{avg}}\left(A(\theta_{\text{mal}}) \cup U_i, \theta_i\right)$
11:         **else**
12:           $\theta_{i+1} = F_{\text{avg}}\left(U_i, \theta_i\right)$
13:     Differentiate the $k - th$ step w.r.t to $\theta_{\text{mal}}$:
14:     $g_{\theta_{mal}} = \nabla_{\theta_{mal}}\left[\sum_{i=1}^{k} \mathcal{L}_{\text{adv}}(D_b, \theta_i) + \mathcal{L}(D_c, \theta_i)\right]$
15:     Update $\theta_{\text{mal}}$ based on $g_{\theta_{mal}}$
16: **return** $\theta_{mal}$

---

with a replaced average over only the subset of modeled users. Over multiple steps $m'$ over which the attacker optimizes the malicious update $\theta_{\text{mal}}$, random data is sampled in every step. We summarize all steps in Algorithm 1. Although the estimation of the adversarial gradient is randomized and based on the distribution of the attacker's data, we find that this scheme is able to reliably generate malicious updates that lead to robuster backdoors.

## 5 EXPERIMENTS

In this section, we thoroughly analyze our attack on three different datasets: CIFAR-10 (image classification) (Krizhevsky et al., 2009), Reddit (next-word prediction) (Caldas et al., 2018), and Sentiment140 (sentiment analysis) (Go et al., 2009). The Reddit dataset naturally contains non-IID partitions. For CIFAR-10 we include results for both IID and non-IID partitions of the dataset. Overall, we show that the proposed method does outperform the baseline of Bagdasaryan et al. (2019) under all tasks and scenarios.

### 5.1 EXPERIMENTATION DETAILS

As described, our experiments implement *fedAVG* (McMahan et al., 2017a; Wang et al., 2021b) with norm-bounding (Sun et al., 2019a). We implement the implicit objective defined in Equation (7) using `functorch` (He & Zou, 2021). For each $\theta_{i,j}$, we sample data points from private clean data randomly. For example, for CIFAR-10 and a batch size of $64$, this still allows us to fit $k = 5$ steps with 10 modeled users onto 11GB of GPU memory. Note that the number of actual users is significantly larger.

For all three datasets, we follow the overall settings discussed in Bagdasaryan et al. (2019); Wang et al. (2020; 2021b). We also randomly split CIFAR-10 over users to make an IID CIFAR-10 task. Details of how each dataset is processed are described below:

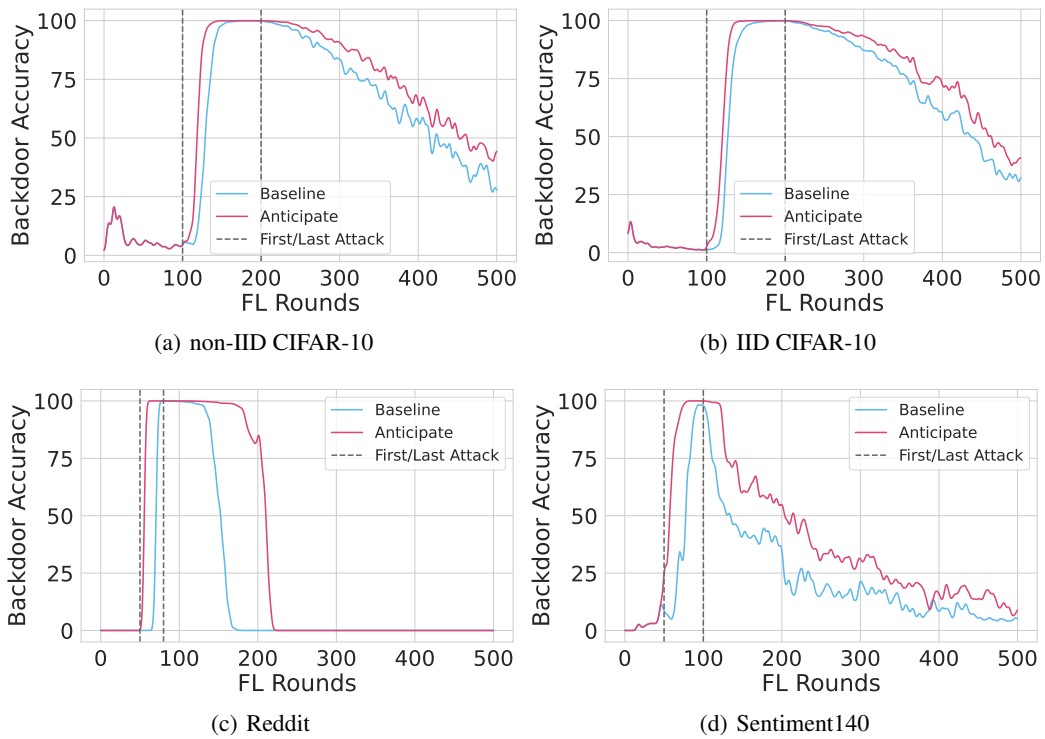

Figure 2: **Attacks over sequential rounds.** In this scenario, an attacker is able to continuously send malicious updates for 100 (a), 100 (b), 30 (c), or 50 (d) rounds. All plots show the run with the first random seed. The attack strictly outperforms previous work under this simpler threat model.

**CIFAR-10**: For CIFAR-10 we investigate an IID partition of data to users and the non-IID split computed through Dirichlet sampling with $\alpha = 1$ (Hsu et al., 2019) both with total 100 users.

For both CIFAR-10 partitions, we choose the backdoor pattern trigger from Gu et al. (2017). The attacker can hence overlay the backdoor pattern on clean data inputs to generate backdoored inputs $D_b$. We choose the label 8 (ship) as the target class for all experiments. For fair comparisons to prior work (Bagdasaryan et al., 2019), we also choose ResNet18 (He et al., 2016). However, it is unclear how to realistically implement norm-bounding for the running stats of Batch Normalization, and global batch norm would typically not be available in a federated system (Ioffe & Szegedy, 2015; Li et al., 2021). Therefore, following Wang et al. (2021b), we replace Batch Normalization with Group Normalization with $G = 32$ (Wu & He, 2018), and $C = 0.5$ for norm-bounding. Empirically, we choose to anticipate $k = 9$ training steps.

**Reddit**: For the Reddit dataset, we take a subset of 2000 users. For next-word prediction, an attacker wants to provide a target word recommendation for users following a trigger sentence. We return to the trigger and target evaluated in Bagdasaryan et al. (2019); the attacker backdoors data by appending `pasta from Astoria is` to the end of a sentence, and the target is to predict the next word `delicious`. Following Bagdasaryan et al. (2019), the adversarial loss on the model output is only computed based on the last word, `is`, of the autoregressive loss. Meanwhile, we re-use the modified 3-layer Transformer model discussed for FL in Wang et al. (2021b) with norm-bounding $C = 1$. We again anticipate $k = 3$ steps.

**Sentiment140**: In Sentiment140 experiments, there are 1000 users in total, and we consider the edge-case examples from Wang et al. (2020) as backdoored data. For example, positive tweets containing `Yorgos Lanthimos` are labeled as negative tweets. For this dataset, we adopt the smaller 3-layer Transformer as above (Vaswani et al., 2017), where the hidden dimension is 1024, and we anticipate $k = 2$ steps. For Sentiment140, we use $C = 0.3$ for norm-bounding.

## 5.2 METRICS

An attacker's goal is to ensure that the backdoor attack accuracy is as high as possible for the global model at the final stage of federated learning. In real scenarios, the number of rounds in which the attacker will be queried by the server and the total number of rounds are unknown to the attacker. For that reason, an attacker wants the attack to be easily implanted and to remain functional as long as possible. Therefore, to test the efficiency of the method, we track the average backdoor accuracy after the first attack ($a_{\text{first}}$) and the average backdoor accuracy after the last attack ($a_{\text{last}}$).

## 5.3 SEQUENTIAL ROUNDS

We first verify that the proposed attack is always an improvement over Bagdasaryan et al. (2019) in the simple setting in which the attacker is queried in all rounds (Wang et al., 2020). We consider 30, 50, and 100 rounds for CIFAR-10, Reddit, and Sentiment140 respectively. These numbers are chosen so that the baseline of Bagdasaryan et al. (2019) can

Table 1: **Results for sequential rounds of attack.** We report average $a_{\text{first}}$ and average $a_{\text{last}}$ of five runs for every experiment. Task 1, task 2, task 3, and task 4 refer to CIFAR-10, IID CIFAR-10, Reddit, and Sentiment140.

|  | Method | Task 1 | Task 2 | Task 3 | Task 4 |
|---|---|---|---|---|---|
| $a_{\text{first}}$ | baseline | 64.72 | 69.72 | 12.22 | 31.87 |
|  | ours | 67.27 | 75.09 | 29.38 | 42.70 |
| $a_{\text{last}}$ | baseline | 63.70 | 70.99 | 11.25 | 27.93 |
|  | ours | 68.04 | 77.82 | 27.82 | 36.92 |

reach a peak backdoor accuracy of at least 95%. For each task, we then repeat the experiment with 5 random seeds and show the plot of the first random seed (to avoid cherry-picking) in Figure 2. We see form these curves that the proposed `anticipate` strategy reaches a peak accuracy slightly faster than the baseline. Especially for the two NLP tasks, when `anticipate` reaches full backdoor accuracy, the baseline's backdoor accuracy is still below 50%. After the last attack, `anticipate` still (slightly) outperforms the baseline across all tasks. In addition, we report average $a_{\text{first}}$ and average $a_{\text{last}}$ of five runs for every experiment in Table 1. Note that in this simple setting the attacker can modify every step of FedAvg, and so there is little risk of adversarial updates fading away. In the next section, we will evaluate the baseline and proposed attacks in the more realistic setting in which a user is queried sporadically, which is the main goal of this paper.

## 5.4 RANDOM ROUNDS

In a real federated learning scenario, it is rare that an attacker is selected for a large number of consecutive rounds, and we will now switch to the more challenging but realistic scenario of random selections. In such scenario, an attacker is randomly selected by the server and does not have any knowledge of the next selected round. This means sometimes there might be a larger time gap between two consecutive attacks. For a fair

Table 2: **Quantitative results for random rounds attack.** We report average $a_{\text{first}}$ and average $a_{\text{last}}$ of five runs for every experiment. Task 1, task 2, task 3, and task 4 refer to non-IID CIFAR-10, IID CIFAR-10, Reddit, and Sentiment140.

|  | Method | Task 1 | Task 2 | Task 3 | Task 4 |
|---|---|---|---|---|---|
| $a_{\text{first}}$ | baseline | 47.38 | 57.05 | 14.04 | 37.85 |
|  | ours | 65.76 | 72.28 | 36.76 | 50.50 |
| $a_{\text{last}}$ | baseline | 47.55 | 59.41 | 10.97 | 26.88 |
|  | ours | 73.75 | 80.61 | 31.26 | 29.93 |

comparison, we randomly select 100 rounds for CIFAR-10, 50 rounds for Reddit, and 100 rounds for Sentiment140 from the first 500 rounds of the whole federated learning routine (to simulate a limited time window for the attack). Overall, these 100/50/100 *malicious updates are only a small fraction of the 5000 overall updates* contributed to the model within the time window of the attack, and an even smaller fraction when compared to 20000 total contributions over the entire 2000 rounds of training. As above, we choose these numbers to yield some success for the baseline attack. Figure 3 shows the plots of each experiment (again from the first random seed). Compared to the experiments of sequential rounds, these more realistic evaluations show that the proposed `anticipate` strategy is significantly more effective than the baseline at attacking the central model. For example, for the non-IID CIFAR-10 experiment, the baseline only maintains a backdoor accuracy of 25% after the attack window, yet `anticipate` maintains backdoor accuracy around 65%. We again include quantitative results in Table 2, computing average $a_{\text{first}}$ and average $a_{\text{last}}$ over the 5 runs of each task.

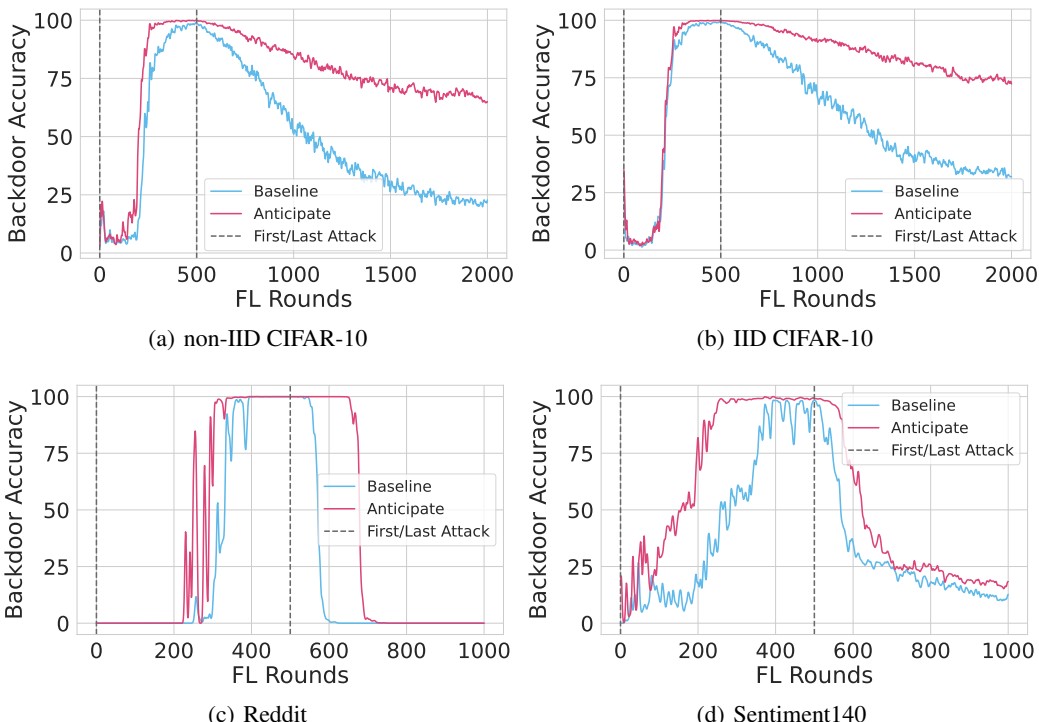

Figure 3: **Attacks over random rounds.** Attacks happen at random 100 (a), 100 (b), 50 (c), or 100 (d) rounds of the first 500 of FL training for four tasks, respectively. All plots show the run with the first random seed. The proposed attack strategy is notably effective under this more realistic threat model.

## 5.5 COMPARISON WITH NEUROTOXIN

We further compare `anticipate` with a recent state-of-the-art attack `Neurotoxin` (Zhang et al., 2022) in the random rounds setting. Briefly, `Neurotoxin` masks out parameters with top-$k\%$ magnitudes. In their setting, they create the mask based on the benign update from the last round to increase the durability of the attack, but it is unlikely to receive an update from the previous round in the random rounds setting, so we use the attacker's clean data to estimate the parameter magnitudes. From the Figure 4, `Neurotoxin` increases the durability on the baseline, but it is slow at injecting the attack. Meanwhile, the overall performance of `Neurotoxin` is behind `anticipate`. We also try to combine `anticipate` with `Neurotoxin`, but unfortunately, `Neurotoxin` slows down `anticipate` a lot, though they end with a close position.

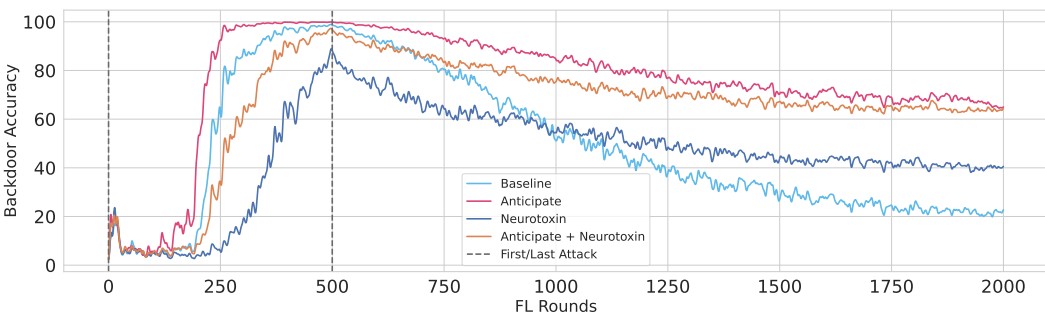

Figure 4: **Comparison with `Neurotoxin`.** Simulating up to 11 `anticipate` steps improves the backdoor persistence.

### 5.6 HYPERPARAMETER STUDY: NUMBER OF STEPS

A central hyperparameter to the attack is the amount of steps to anticipate (and subsequently to evaluate the objective on). In Figure 5, we compare anticipation intervals between 3 and 11 on the random rounds attack for non-IID CIFAR-10. We find that the larger the number of steps an attacker employs, the faster the attack is implanted. However, interestingly, such quick implantation does not necessarily mean that the attack lasts longer. The highest accuracy at the end of training is actually reached at $k = 9$ steps. However, any number of steps greater than 3 improves over the greedy baseline attack which corresponds to $k = 0$.

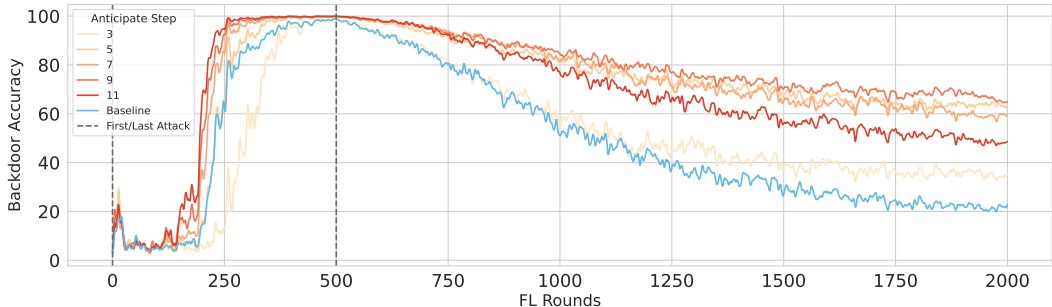

Figure 5: **Backdoor accuracy among different `anticipate` steps.** Simulating up to 11 `anticipate` steps improves the backdoor persistence.

### 5.7 EFFICIENT ANTICIPATE

We notice that directly optimizing all $k$ steps can be time-consuming and computationally heavy. Therefore, we investigate three different variations of `anticipate` to boost the efficiency of the algorithm: A) the attacker only calculates the loss on the last step. B) for each iteration, the attacker randomly chooses a number $k' \leq k$ and calculates the loss on all $k'$ steps. C) the attacker follows the variant B, but only calculates the loss on step $k'$. In Figure 6, we plot the performance and average optimization time ($k = 9$) of three vari-

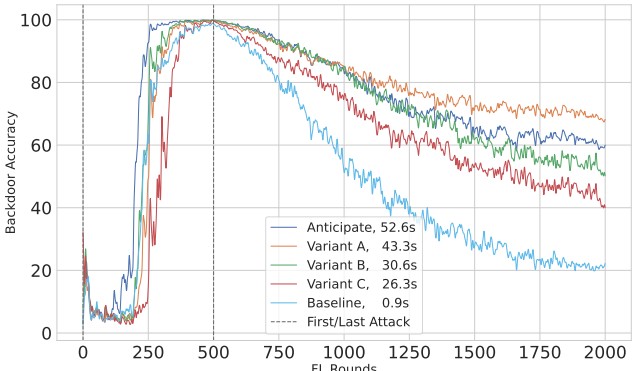

Figure 6: **Efficient Variants of `anticipate`**. More efficient and fast `anticipate` can still improve the baseline.

ants. Variant A gives the most durable attack at the end due to it directly targets the furthest step $k$. However, variant A is much slower than the normal `anticipate` during the injecting time. We can reduce a huge amount of optimization time by choosing variant C, and it is still better than the baseline. Meanwhile, this strategy uses less computing resources than the original `anticipate`. Although, all variants are much slower than the baseline, in a real scenario, we believe that such a delay would be hidden in the general update delay that is common in federated learning due to communication issues and hardware heterogeneity, especially for mobile devices as described in Bonawitz et al. (2019). The attacker can further compute their update within the same time budget by using hardware that is several times stronger than the slowest hardware allowed by the server.

## 6 CONCLUSION

We evaluate the feasiblity of backdoor attacks in the realistic regime where users are numerous and *not consistently queried* by the central server. We do this by considering an attack that models simulated FedAvg updates and choose adversarial perturbations that are unlikely to be over-written by other users. Through a series of experiments on backdoor attacks for image classification, next-word prediction, and sentiment analysis, we show that this strategy leads to strong backdoor attacks, even in scenarios where the attacker has relatively few opportunities to influence the model.

ETHICS STATEMENT - MITIGATIONS

Backdoor attacks have the theoretical potential to be used to disrupt federated learning system used in various applications, for example, for application systems described in (Bonawitz et al., 2019; Paulik et al., 2021; Dimitriadis et al., 2022). Aside from disruption to these systems, backdoor attacks have the potential to influence decision makers to favor centralized machine learning systems, which reduces privacy afforded to users of such systems.

Our key message here is that for systems defended by only norm-bounding scenario, the community's estimation of attack capabilities ask suggested by Bagdasaryan et al. (2019); Sun et al. (2019b) underestimates the potential validity of such an attack. Although norm-bounding is a strong and efficient defense against attack evaluated in (Sun et al., 2019b), the attack discussed in this work still works against a tight norm bound and in realistic scenarios with few attack opportunities. This shows that additional defenses should be considered on top of norm-bounding, which by itself continues to be a necessary defense to mitigate the individual influence of an attack that sends extreme malicious updates, like model replacement (Bagdasaryan et al., 2019; Bhagoji et al., 2019). We verify in Appendix A.5, that the attack discussed in this work is specific to norm-bounding as a defense and can be mitigated using other groups of defenses, such as Krum, Multi-krum, or median aggregation, as the attack was constructed to break norm-bounded systems in particular.

REPRODUCIBILITY STATEMENT

We provide code with the supplementary material to reproduce all attacks.

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

# A   APPENDIX

## A.1   ABLATION STUDY: AMOUNT OF PRIVATE DATA

The number of private data points an attacker holds is critical for how well the attacker can estimate the benign user's contribution. Intuitively, the more private data an attacker holds the more accurately the attacker can predict other users. To estimate the effect of data on the attack, we test variations where the attacker holds 100, 300, 500, and 700 data points for non-IID CIFAR-10, and show backdoor evaluations in Figure 7. In this case, all other benign users have 500 images. For the experiment with data size = 0 in the figure, the attacker replaces the data for other users with random noise, using their own 500 images only to create $D_b$. We find that more data does robustify backdoor performance, and that random data is insufficient to model other users. However, even with only 100 data points, the estimation is notably successful.

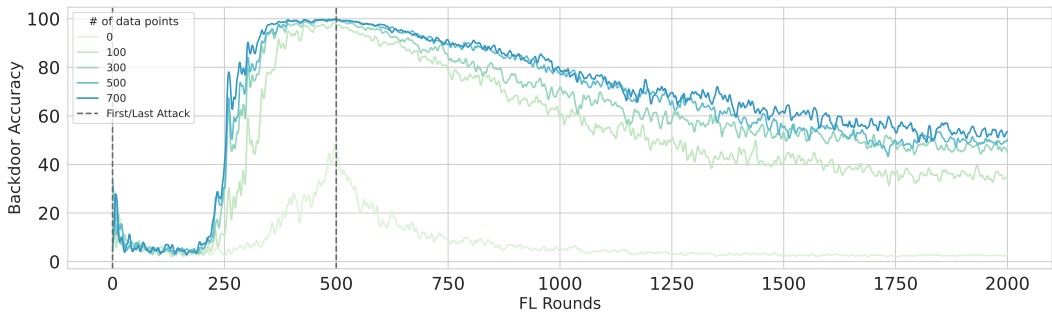

Figure 7: **Backdoor accuracy with different amounts of private data held by the attacker.** Even a limited amount of local data available to the attacker is sufficient for a strong attack.

## A.2   ABLATION STUDY: NUMBER OF USERS PER ROUND

Another important factor in federated learning is the number of users involved in each round. The aggregation between a larger number of users might be more difficult for implanting the attack. We continue to assume that the attacker is sometimes in control of a single user per round. Again, we test 5 cases with 5, 10, 15, 20, and 30 users per round for non-IID CIFAR-10. Figure 8 shows how effective the attack is in different situations. For the case with less than 20 users per round, the attack is still effective with 100 random rounds attack in the first 500 rounds. However, when there are more than 20 users per round, the attack still needs more rounds to work.

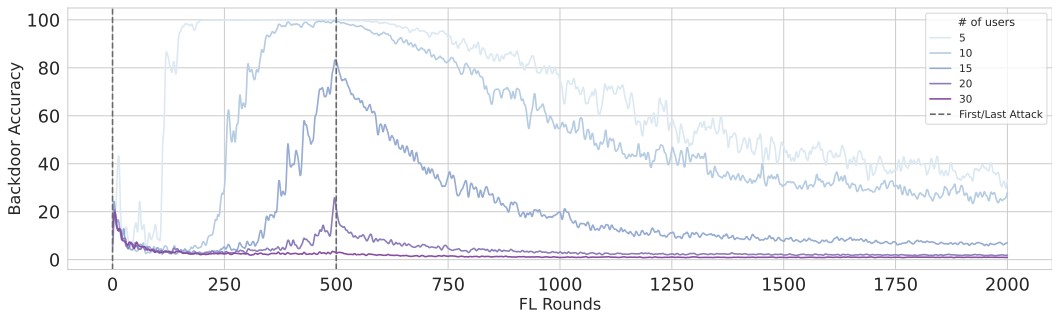

Figure 8: **Backdoor accuracy with different numbers of users per round.** The effectiveness of the attack decreases as the number of users per round increases.

## A.3    ABLATION STUDY: DATA HETEROGENEITY

Figure 9 shows the results on different $\alpha$ for Dirichlet sampling in random round attack setting for non-IID CIFAR-10. The smaller $\alpha$ is, the more non-iid sampling is. The effectiveness of the attack decreases as the data heterogeneity increases. However, Anticipate can still improve the baseline, even $\alpha = 0.25$. According to Wang et al. (2021a), $\alpha = 1$ is a reasonable value for simulating non-iid federated learning.

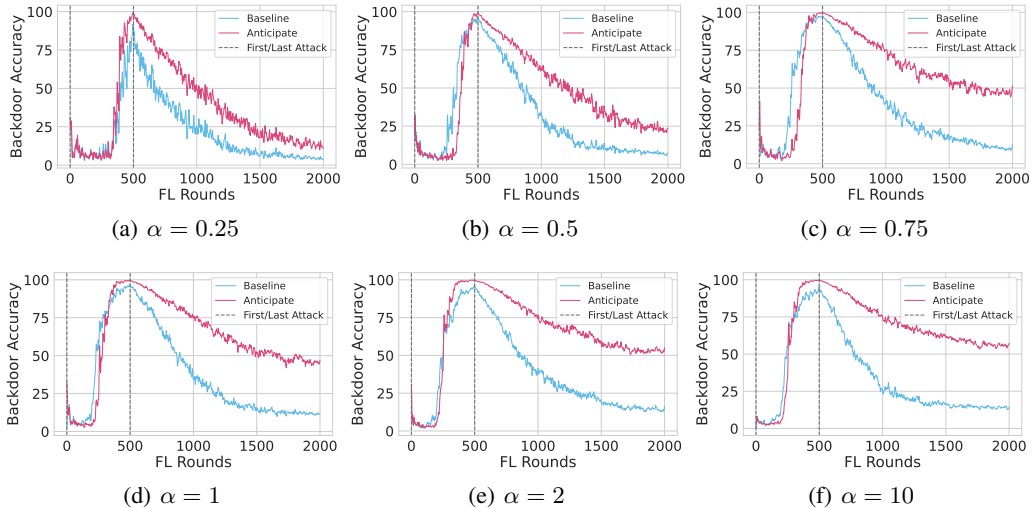

(a) $\alpha = 0.25$          (b) $\alpha = 0.5$          (c) $\alpha = 0.75$

(d) $\alpha = 1$          (e) $\alpha = 2$          (f) $\alpha = 10$

Figure 9: **Backdoor accuracy with different $\alpha$ for Dirichlet sampling.** The effectiveness of the attack decreases as the data heterogeneity increases.

## A.4    MAIN TASK ACCURACY

Figure 10 reports the main task accuracy during the training. There is no significant difference between the baseline and Anticipate.

## A.5    ATTACKS AGAINST STRONG CLUSTERING DEFENSES

Instead of norm-bounding, we also implement four defenses from the literature: Krum (Blanchard et al., 2017), Multi-krum (Blanchard et al., 2017), Median (Yin et al., 2018), and Trimmed-mean (Yin et al., 2018). Krum and Multi-krum are vector-wise abnormal detection defenses, whereas Median and Trimmed-mean are element-wise abnormal detection defenses, but note that *the proposed attack is not designed to break these defenses*. As shown in Figure 11, we can verify that this is indeed the case and Krum, Multi-krum, and median play very effective roles in defending against both the baseline attack of Bagdasaryan et al. (2019) and Anticipate. Main task accuracy drops noticeably for all of them in Figure 12, illustrating why their widespread adoption in applications might be so far reluctant. Interestingly, Trimmed-mean both doesn't reduce main task accuracy as much, and is the defenses where the attack works out of the box, although the attack's durability is reduced. We argue that more advanced defenses such as these should find more use in modern FL applications so that server owners can choose strong defenses and balance the trade-off between the effectiveness of the defense and the drop in performance.

## A.6    OBJECTIVE WEIGHTS

During the implementation, we average two objects in Equation (7). Now, we modify the loss as:

$$\theta^* = \underset{\theta_{\mathrm{mal}}}{\arg\min} \sum_{i=1}^{k} \alpha \mathcal{L}_{\mathrm{adv}}(D_b, \theta_i(\theta_{\mathrm{mal}})) + (1 - \alpha)\mathcal{L}(D_c, \theta_i(\theta_{\mathrm{mal}})). \qquad (8)$$

As shown in Figure 13, if $\alpha$ is greater than or equal to 0.5, there is no significant difference.

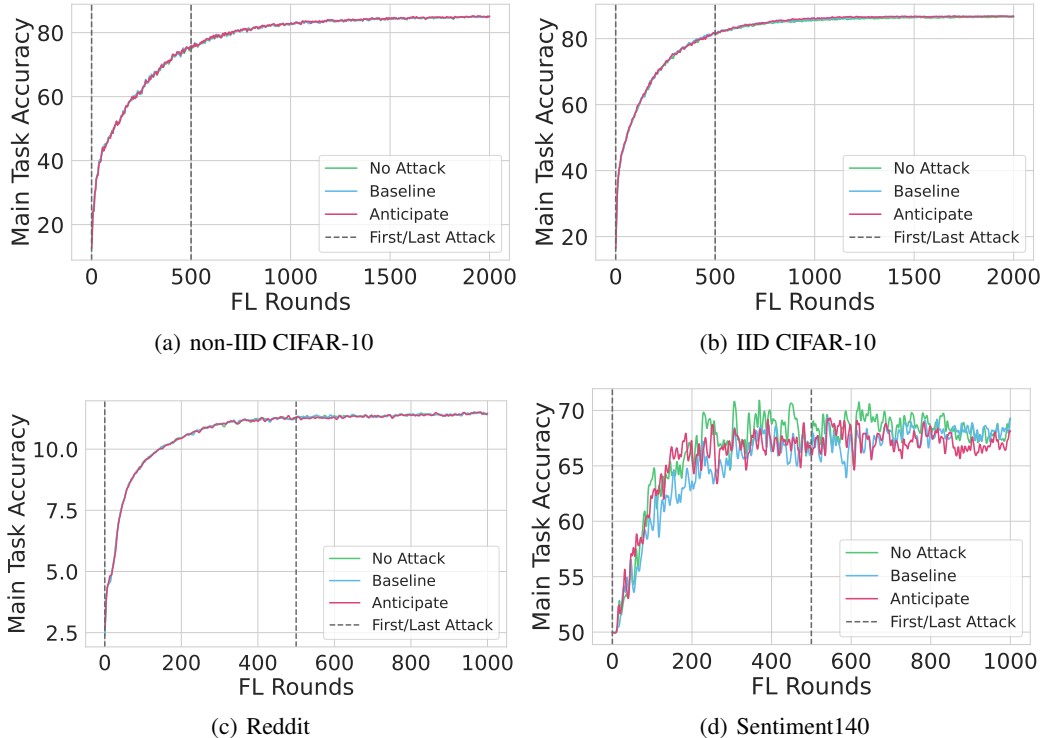

Figure 10: **Main task accuracy.**

## A.7    ABLATION STUDY: NORM-BOUNDING

We show the results with different norm-bounding thresholds in Figure 14.

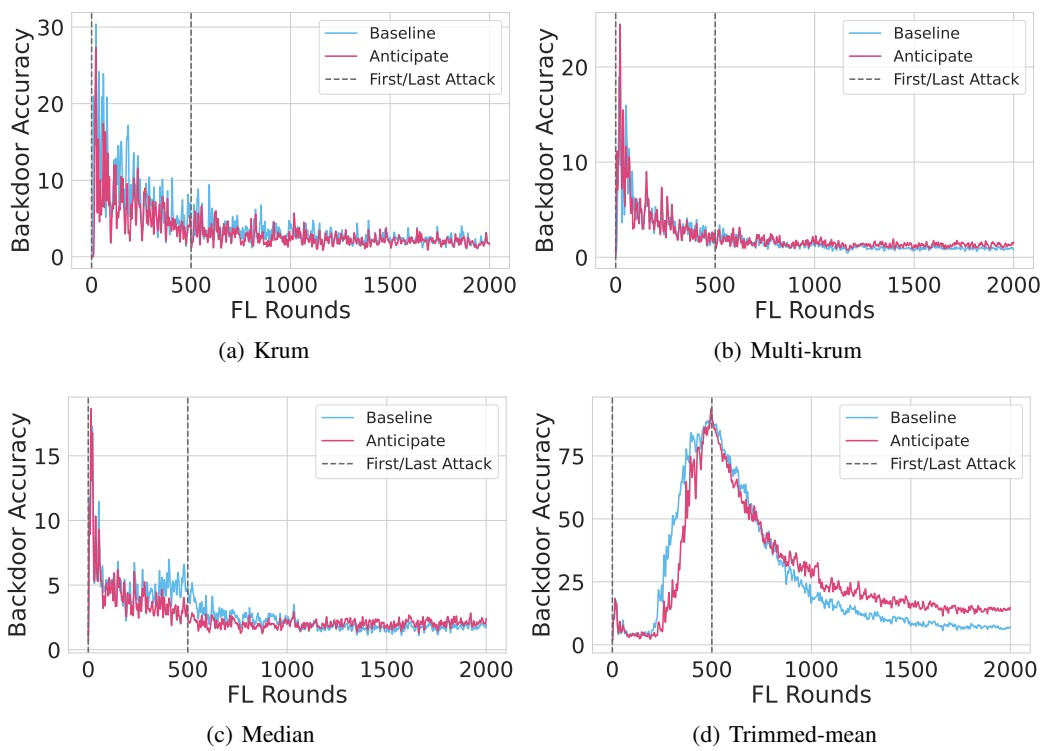

Figure 11: **Backdoor accuracy under different defenses.**

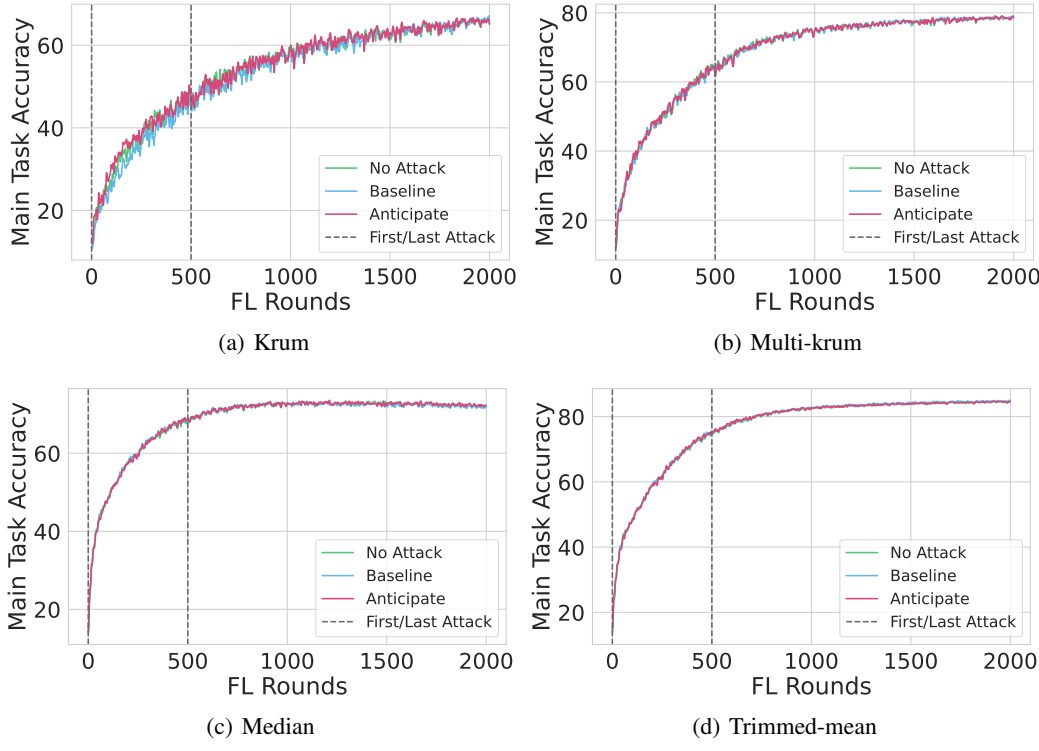

Figure 12: **Main task accuracy under different defenses.**

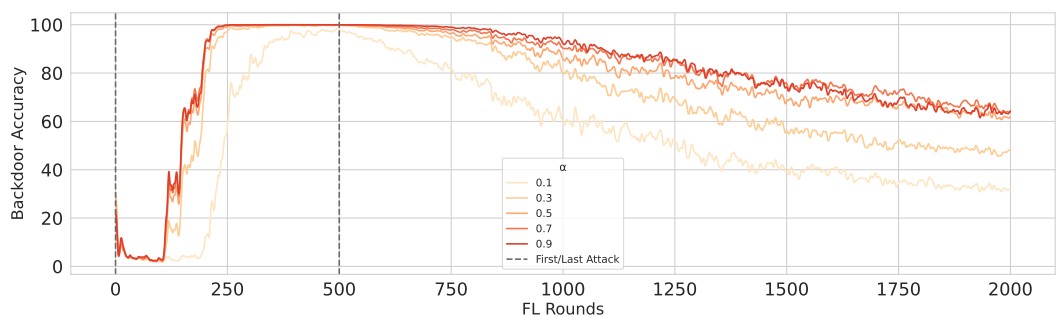

Figure 13: **Backdoor accuracy with different loss weights.**

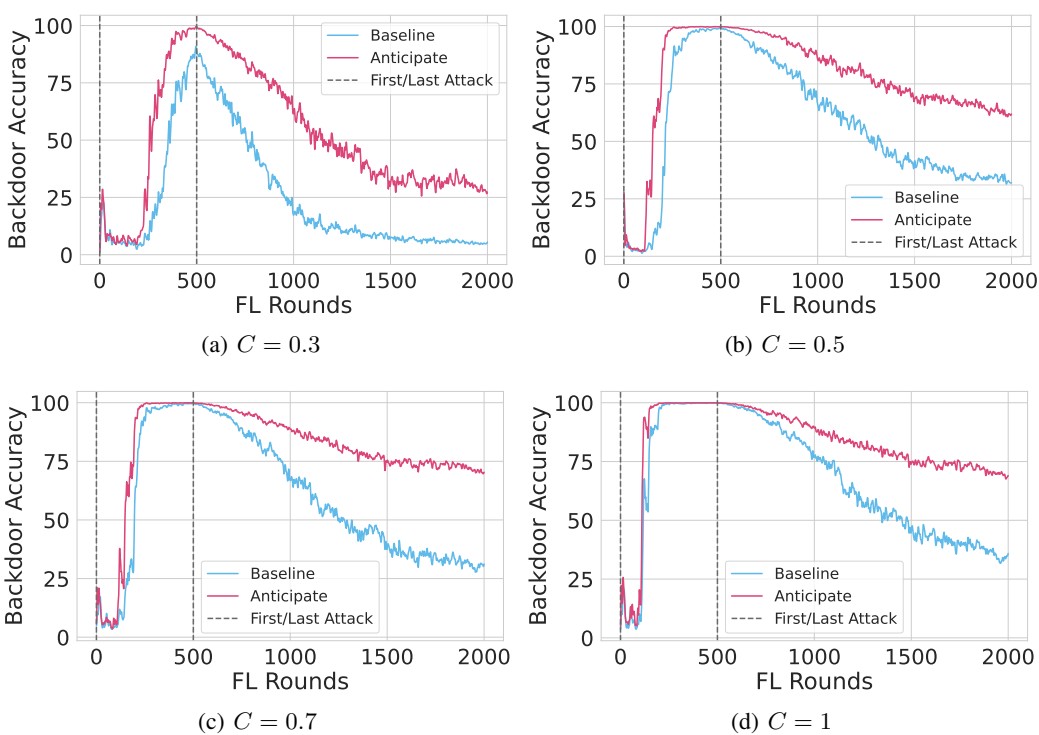

Figure 14: **Different Norm-bounding Thresholds.**

