# OpenReview forum: "Thinking Two Moves Ahead: Anticipating Other Users Improves Backdoor Attacks in Federated Learning"
_ICLR.cc/2023/Conference — Submitted to ICLR 2023_

### Official Review · Reviewer_xjs7 · 2022-10-24

**Confidence:** 5
**Correctness:** 3
**Technical Novelty And Significance:** 2
**Empirical Novelty And Significance:** 2
**Recommendation:** 3

**Clarity, Quality, Novelty And Reproducibility:**

Algorithm 1 needs to be clarified in a couple of places. Does the group of users U_i change over time? How is the number of simulated benign users n' determined? Also, the algorithm uses FedAvg to simulate the server's aggregation rule. Does it include the norm clipping step?

In terms of novelty, the idea of simulating the behavior of benign agents in federated learning has been considered in recent work, e.g., [1]. Although the focus in [1] was on model poisoning, the idea of building a model of the FL system and benign agents using common knowledge and the attacker's local data can be readily applied to backdoor attacks. Further, [1] used reinforcement learning to obtain an attack policy that optimizes a long-term objective, which is more general and effective than the myopic approach adopted in this paper.

[1] Wen Shen, Henger Li, and Zizhan Zheng. Learning to Attack Distributionally Robust Federated Learning. NeurIPS-20 Workshop on Scalability, Privacy, and Security in Federated Learning (SpicyFL).






**Strength And Weaknesses:**

Strengths

1. The idea of simulating benign users' behavior to develop more advanced attacks against federated learning is interesting.
2. The proposed backdoor attack is effective against the norm-bounding defense.

Weaknesses

1. The paper considers an oversimplified setting with a single malicious device, and the simulation algorithm only considers a single attack step (the current step) and ignores the possibility that the device may be sampled again soon. Further, the algorithm simulates a fixed number of steps before the next attack happens, which is inaccurate given the randomness of the sampling algorithm used by the server. As the attacker has no access to other users' local data, the paper simply simulates their behavior from the attacker's local data, which can also be inaccurate for non-iid local data distributions.
2. Because of the simplified design, the proposed attack can only break the norm-bounding defense and completely fails when the server adopts Krum, Multi-Krum, or Median, as shown in the appendix. Note that these are not really "advanced" defenses as they are not designed for backdoor attacks. Recent defenses, especially detection-based and post-training-based defenses, obtain more promising results against backdoors, which are completely ignored in the paper.
3. For the norm-bounding defense considered in the paper, the norm threshold C is crucial for achieving a good tradeoff between the main task accuracy and robustness against attacks. But I cannot find the value of C used in the experiments, and there is no ablation study that investigates the impact of C on main task accuracy and backdoor accuracy.

**Summary Of The Paper:**

A new backdoor attack against federated learning is proposed. The paper considers a single malicious device that can share malicious updates with the server to inject a backdoor into the centralized model, and the server adopts FedAvg with norm clipping as the aggregation rule. The main idea is to account for benign users' behavior in future rounds to make the malicious updates more persistent. Simulation results demonstrate the effectiveness of the proposed attack against the norm-bounding defense.

**Summary Of The Review:**

The idea of simulating benign users' behavior to develop more advanced attacks against federated learning is interesting. However, the paper considers an over-simplified setting, and the proposed backdoor attack is only effective against the norm-bounding defense and completely fails against other common defenses. The proposed solution is far from being sophisticated.

---

> ### Author Response · Authors · 2022-11-19
> **Response**
>
> Thank you for your feedback and we appreciate your time in reviewing this submission. Below, we address specific questions that you brought up:
>
> > The paper considers an oversimplified setting with a single malicious device, and the simulation algorithm only considers a single attack step (the current step) and ignores the possibility that the device may be sampled again soon. Further, the algorithm simulates a fixed number of steps before the next attack happens, which is inaccurate given the randomness of the sampling algorithm used by the server. As the attacker has no access to other users' local data, the paper simply simulates their behavior from the attacker's local data, which can also be inaccurate for non-iid local data distributions.
>
> We want to clarify several points in your summary. We believe our setting is more realistic settings with multiple attackers per round, because attacks are rare, a single attacker is unlikely to be called again soon. Our settings can be thought to generalize to multiple attackers, but each called rarely. We would argue that this is not oversimplified, but overly realistic. We simulate a fixed number of benign update rounds, not future attacks. Further, we extensively test non-iid local data to verify the effectiveness of the proposed attack.
>
> > More advanced backdoor defense
>
> In this paper, we want to highlight the effectiveness of backdoor attacks against defenses deployed in real-world scenarios. We agree with you that a conclusion of our work, in contrast to previous papers (such as Sun et. al, 2019, who find norm-bounding to be sufficient) is that stronger defenses are necessary. We believe this work to be an important example for the community that underscores the value of new defenses and calls for their adoption in deployed systems.
>
> Sun, Z., Kairouz, P., Suresh, A. T., & McMahan, H. B. (2019). Can you really backdoor federated learning?. arXiv preprint arXiv:1911.07963
>
> > Norm-bounding
>
> Thanks for bringing this point up. We added extensive experiments to the appendix, where we cover additional values for all norm-bounding thresholds in the experiment part in the new version. In the main body, we use $C = 0.5, 1, 0.3$ for  CIFAR-10, Reddit, and Sentiment140 respectively, but now include many variations.
>
> We added these ablation experiments on norm-bounding thresholds in Appendix A.7 and Fig.14. As we can see, Anticipate can consistently improve the baseline across all norm-bounding thresholds.
>
> > Algorithm 1 needs to be clarified in a couple of places. Does the group of users U_i change over time? How is the number of simulated benign users n' determined? Also, the algorithm uses FedAvg to simulate the server's aggregation rule. Does it include the norm clipping step?
>
> We randomly sample a batch of $b$ data points from $D_c$, for each $U_i$, they are changing. By default, the number of users per round is known information to the users. We didn't include norm clipping for the current implementation. Norm-clipping is not differentiable, so our simulation doesn't include it.
>
> > In terms of novelty, the idea of simulating the behavior of benign agents in federated learning has been considered in recent work, e.g., [1].
>
> Thanks for pointing out this. We added it to the related work. In comparison, our work looks at a more realistic scenario.
>
> Thanks for your time. We hope our clarifications answer your questions.

---

### Official Review · Reviewer_VoJT · 2022-10-25

**Confidence:** 3
**Clarity, Quality, Novelty And Reproducibility:** Difficult to read, and hard to reprod…
**Correctness:** 3
**Technical Novelty And Significance:** 3
**Empirical Novelty And Significance:** 2
**Recommendation:** 5

**Strength And Weaknesses:**

1. The paper claims two moves ahead would be better than other attack methods such as norm-bounding. I guess that Eq. (4) with two loss functions would be the two moves, but the parameter between the moves is not given. Generally, we need to add a parameter to compromise the two loss functions in eq.(4), and add more experiments to show the results with different "move weights". However, it seems the moves are averaged and the paper does not explain why the average move is effective and why the compromise weight is not required as expected.

2. The motivation and experiments are with strong control. It would be persuasive to add real-world motivation and experimental data.  Plus, I found that a demo version of this paper has been published in ICML. Because a previous version of this paper has already published, it would be better to add discussions about the difference between the two papers.



**Summary Of The Paper:**

The paper proposes an attack that anticipates and accounts for the entire federated learning pipeline, including behaviors of other clients, and ensures that backdoors are effective quickly and persist even after multiple rounds of community updates. Experiments on datasets  demonstrate the performance of this attack on image classification, next-word prediction, and sentiment analysis.



**Summary Of The Review:**

Please see the above.

---

> ### Author Response · Authors · 2022-11-19
> **Response**
>
> We appreciate the viewer's time in giving us constructive suggestions. Here we respond to the individual questions.
>
> > Generally, we need to add a parameter to compromise the two loss functions in eq.(4) and add more experiments to show the results with different "move weights".
>
> Our previous implementation averages two losses. However, in our updated draft, we now include a hyperparameter $\alpha$ as the "move weight," so eq.(4) (now eq.7 in the new version) is modified as:
> $\theta^* = \arg\min_{\theta_\text{mal}} \sum_{i=1}^{k} \alpha\mathcal{L}_\text{adv}(D_b, \theta_i(\theta_\text{mal})) + (1-\alpha)\mathcal{L}(D_c, \theta_i(\theta_\text{mal}))$.
>
> We conduct extensive experiments in Appendix A.6 and Fig.13. We find that if $\alpha$ is below 0.5, the effectiveness of the baseline attack reduces, but if $\alpha$ is above $0.5$, there is no gain.
>
> > The motivation and experiments are with strong control. It would be persuasive to add real-world motivation and experimental data. Plus, I found that a demo version of this paper has been published in ICML. Because a previous version of this paper has already been published, it would be better to add discussions about the difference between the two papers.
>
> Please note that this work has not been published before. This attack has been discussed at a previous non-archival ICML workshop in a poster format.
>
> Thanks for your feedback. We hope that the additional experiments above elucidate the questions about $\alpha$.

---

### Official Review · Reviewer_Fbk1 · 2022-11-03

**Confidence:** 4
**Correctness:** 3
**Technical Novelty And Significance:** 2
**Empirical Novelty And Significance:** 2
**Recommendation:** 5

**Clarity, Quality, Novelty And Reproducibility:**

Poor quality.
Nice clarity.
Poor originality.

**Strength And Weaknesses:**

Strengths:

(a) This paper is well-written and easy to read.

(b) I appreciate that extensive experiments are provided in this work.

Weaknesses:

(a) Should m steps be represented in Eq.(1)? If yes, please revise Eq.(1) and provide some references. If no, please provide a reasonable explanation.

(b) In Section 3, it is mentioned that ``We believe this threat model with random attack opportunities is a natural step towards the evaluation of risks caused by backdoor attacks in more realistic systems''. Please provide a proper example to demonstrate the application in reality.

(c) In Section 4, some formulas are not numbered.

(d) It is hard to imagine how to implement the method proposed in this work. In the lines 9 to 12 of Algorithm 1, the proposed method needs to calculate ($\theta_1, \theta_2, ..., \theta_k$). It is ok to calculate these parameters, but it is hard to imagine how to calculate line 14. In line 14, it requires to differentiate k-th step w.r.t $\theta_{mal}$. Recall the calculation of $\theta_{u,j}$ (such as Eq.(1)), we can find that it needs to calculate the derivatives. These mean that there are k-th partial derivatives in the calculation of line 14, which are very difficult to implement. This point is my main concern. Please provide a very detailed explanation and describe how to calculate it.

**Summary Of The Paper:**

This work focuses on the model poisoning and backdoor attacks in federated learning. It proposes an attack that anticipates and accounts for the entire federated learning pipeline, including behaviors of other clients.

The main contributions can be summarized as:

(1) Proposing a backdoor attack in federated learning (anticipate).

(2) Providing extensive experiments.

**Summary Of The Review:**

I recommend marginally below the acceptance threshold. I will give my final score based on the response and other reviewer's comments.

---

> ### Author Response · Authors · 2022-11-19
> **Response**
>
> Thank you for your feedback and we appreciate your time in reviewing this submission. Below, we address specific questions that you brought up:
>
> > Should m steps be represented in Eq.(1)? If yes, please revise Eq.(1) and provide some references. If no, please provide a reasonable explanation.
>
> Thanks for pointing this out. We have clarified the notation of Eq.(1).
>
> > In Section 3, it is mentioned that ``We believe this threat model with random attack opportunities is a natural step towards the evaluation of risks caused by backdoor attacks in more realistic systems''. Please provide a proper example to demonstrate the application in reality.
>
> During the training, the server randomly samples users from a large pool, so the round that the attacker has been selected is random. However, in the case of the sequential round, the attacker assumes they are consecutively been selected by the server in a time window. Therefore, we believe the random round setting is more realistic/difficult for the attacker.
>
> > In Section 4, some formulas are not numbered.
>
> Thanks for bringing this up. We have labeled all formulas in the new draft.
>
> > It is hard to imagine how to implement the method proposed in this work.
>
> We implemented our algorithm using Functorch(https://github.com/pytorch/functorch). Functorch allows us to directly do differentiable operations on model parameters. For example, to achieve fedAvg, we can just easily sum all simulated users' model parameters and divide the number of users. Meanwhile doing fedAvg, Functorch is saving all operations in the gradient graph. Another important functionality of Functorch is that it allows us to use the "grad" function to simulate users' normal updates and include them in the gradient graph. After the simulation, we can simply do a normal "loss.backward()" to get gradients for the attacker's parameters and do the update. In summary, line 14 can be simply implemented using automatic differentiation tools. You may check the detailed implementation in `anticipate_algo.py` file in our supplementary material.
>
> Thank you for your feedback on our submission. We hope to have clarified all questions raised in your review in our response and our updated draft.

---

### Decision · Program_Chairs · 2023-01-20

**Decision:**

Reject

**Justification For Why Not Higher Score:**

N/A

**Justification For Why Not Lower Score:**

N/A

**Metareview: Summary, Strengths And Weaknesses:**

The paper presents a backdoor attack in federated learning and extensive experiments show the effectiveness of the proposed method. But reviewers find that the quality and presentation of this paper is not good at all. The novelty is very limited.